**Subject Category:**
Biology (whole organism)

ecology/evolution/taxonomy and systematics

chemosynthetic ecosystems, *Desbruyeresia*, Mollusca, new species, phylogeny, *Provanna*

**Author for correspondence:**
Chong Chen
e-mail: cchen@jamstec.go.jp

# Four new deep-sea provannid snails (Gastropoda: Abyssochrysoidea) discovered from hydrocarbon seep and hydrothermal vents in Japan

Chong Chen[1], Hiromi Kayama Watanabe[1]
and Takenori Sasaki[2]

[1]X-STAR, Japan Agency for Marine-Earth Science and Technology (JAMSTEC), 2–15 Natsushima-cho, Yokosuka, Kanagawa 237-0061, Japan
[2]The University Museum, The University of Tokyo, 7-3-1 Hongo, Bunkyo-ku, Tokyo 113-0033, Japan

CC, 0000-0002-5035-4021; HKW, 0000-0001-5031-9018

Recently, the species richness of provannid gastropods inhabiting chemosynthetic ecosystems in the northwestern Pacific has been reassessed, revealing a much higher diversity than previously realized. Here, we describe four further new species, two in the genus *Desbruyeresia* and two in the genus *Provanna*. Their generic placement was confirmed by both shell and radula morphology, as well as phylogenetic reconstruction using the mitochondrial cytochrome *c* oxidase subunit I gene. *Desbruyeresia armata* n. sp. from vent fields in the Izu-Ogasawara Arc is characterized by a stout shell with numerous tubercles or short spines and marginal teeth coarsely serrated into only four denticles. *Desbruyeresia costata* n. sp. from Okinawa Trough vents is distinguished from other congeners by an elongate shell with strong axial ribs and obsolete spiral ribs. These represent the first *Desbruyeresia* species from Japanese waters. *Provanna fenestrata* n. sp. discovered from two vent fields in the Okinawa Trough is instantly recognizable from its uniquely regular rectangular lattice sculpture; *Provanna stephanos* n. sp. is a surprising new discovery from the supposedly well-explored 'Off Hatsushima' hydrocarbon seep site in Sagami Bay, and is highly distinctive with two characteristic rows of strongly spinous spiral ribs. The discovery of these new species in relatively well-explored chemosynthetic ecosystems in Japan indicates that the biodiversity of such systems remains poorly documented.

# 1. Introduction

Since the first discovery of hydrothermal vents in the Galápagos Rift in 1977 [1], intensive investigations have revealed hundreds of deep-sea chemosynthetic ecosystems in a variety of forms including not only vents but also methane seeps and organic falls [2]. These ecosystems host lush communities of organisms energized by bacterial chemosynthetic production and contain largely endemic taxa not found in non-chemosynthetic environments, with 71% out of 712 species in hydrothermal vents, for example [3]. The Japanese waters are in the vicinity of four tectonic crust plates converging together, a tectonic setting leading to very active plate motion which fuels more than 50 chemosynthetic sites relatively close to land [4]. This makes accessing chemosynthetic ecosystems in this area relatively straightforward, and numerous research cruises carried out over the last decades have made this area one of the best explored in the world for such ecosystems [5].

Gastropod molluscs are among the most abundant and diverse animal groups in deep-sea chemosynthetic communities [6]. The abyssochrysoid family Provannidae is endemic to these communities around the world, and often dominates in number and biomass [6,7]. The family currently comprises five genera: *Alviniconcha* and *Ifremeria* which are large (over 80 mm in maximum shell height) and house endosymbiotic bacteria in the gill epithelium [8]; *Provanna, Debsruyeresia* and *Cordesia* which are small (less than 20 mm in shell height) deposit grazers [6,9–11]. There are currently six recognized extant species of *Alviniconcha*, a single species of *Ifremeria* [12], 19 extant and seven extinct *Provanna* species [7,11,13,14], six living species of *Desbruyeresia* [15] and a single extant species of *Cordesia* [10]. A further abyssochrysoid genus, *Rubyspira*, is known from whale falls and contains two recent species but has not been assigned to a family with certainty [16].

Until recently, it was considered that the only provannids inhabiting Japanese waters were three species of *Provanna*, including *Provanna glabra* Okutani, Tsuchida & Fujikura, 1992, in methane seeps of Sagami Bay and hydrothermal vents of Okinawa Trough, *P. abyssalis* Okutani & Fujikura, 2002, and *P. shinkaiae* Okutani & Fujikura, 2002, in the methane seeps of Japan Trench [13,14]. Recently, however, it was revealed through combined morphological and molecular analyses that *P. glabra* is actually restricted to Sagami Bay seeps with a very similar sister species, *P. subglabra*, inhabiting Okinawa Trough vents along with three further species [7]. A single specimen of *Alviniconcha adamantis*, Johnson *et al.*, 2014, is also known from Suiyo Seamount in the Izu-Ogasawara (aka. Izu-Bonin) Arc [17], but no further specimens have been discovered so far [12].

Upon examining materials obtained from chemosynthetic ecosystems around Japan, including a hydrocarbon seep and several hydrothermal vents, it became apparent that four additional provannid species new to science exist in Japanese waters. These include two species of *Desbruyeresia* (both from hydrothermal vents) and two species of *Provanna* (one from hydrocarbon seep and one from vents). The two new *Desbruyeresia* species represent the first described species from Japan and the northwestern Pacific. Here, these four novel taxa are characterized and formally described, and new biogeographic insights gained through their discovery are discussed.

# 2. Material and methods

## 2.1. Sample collection

Hydrocarbon seep and hydrothermal vent sites sampled are shown in figure 1 and table 1. One hydrothermal field sampled, the Higa site northeast of Gima Hill in the Okinawa Trough [18], was recently discovered in the Okinawa Trough (also known as the Acoustic Petit Anomaly (APA) site [19]) by detecting acoustic water column anomalies (likely derived from $CO_2$ bubbles discharged from vents), using an EM122 (Kongsberg Maritime, Kongsberg, Norway) MBES system on R/V *YOKOSUKA* (cruise YK15-14, principal scientist: Kentaro Nakamura), as outlined in a previous study [20].

Provannid snails were collected from the chemosynthesis-based ecosystems using suction samplers mounted on the remotely operated vehicle (ROV) *Hyper-Dolphin* on-board cruises NT07-17 (principal scientist: Kokichi Iizasa) and NT11-20 (principal scientist: Jun-ichiro Ishibashi) of R/V *NATSUSHIMA* and KS-16-04 (principal scientist: Akinori Yabuki) of R/V *SHINSEI MARU*; as well as ROV *KAIKO* (with vehicle *Mk-IV*) on-board cruises KR15-16 (principal scientist: Shinsuke Kawagucci) and KR15-17 (principal scientist: Hiroyuki Yamamoto) of R/V *KAIREI*. Upon recovery on board the research vessels, provannid snails were sorted out and immediately preserved in 70% ethanol, 99% ethanol or frozen in a −20°C freezer.

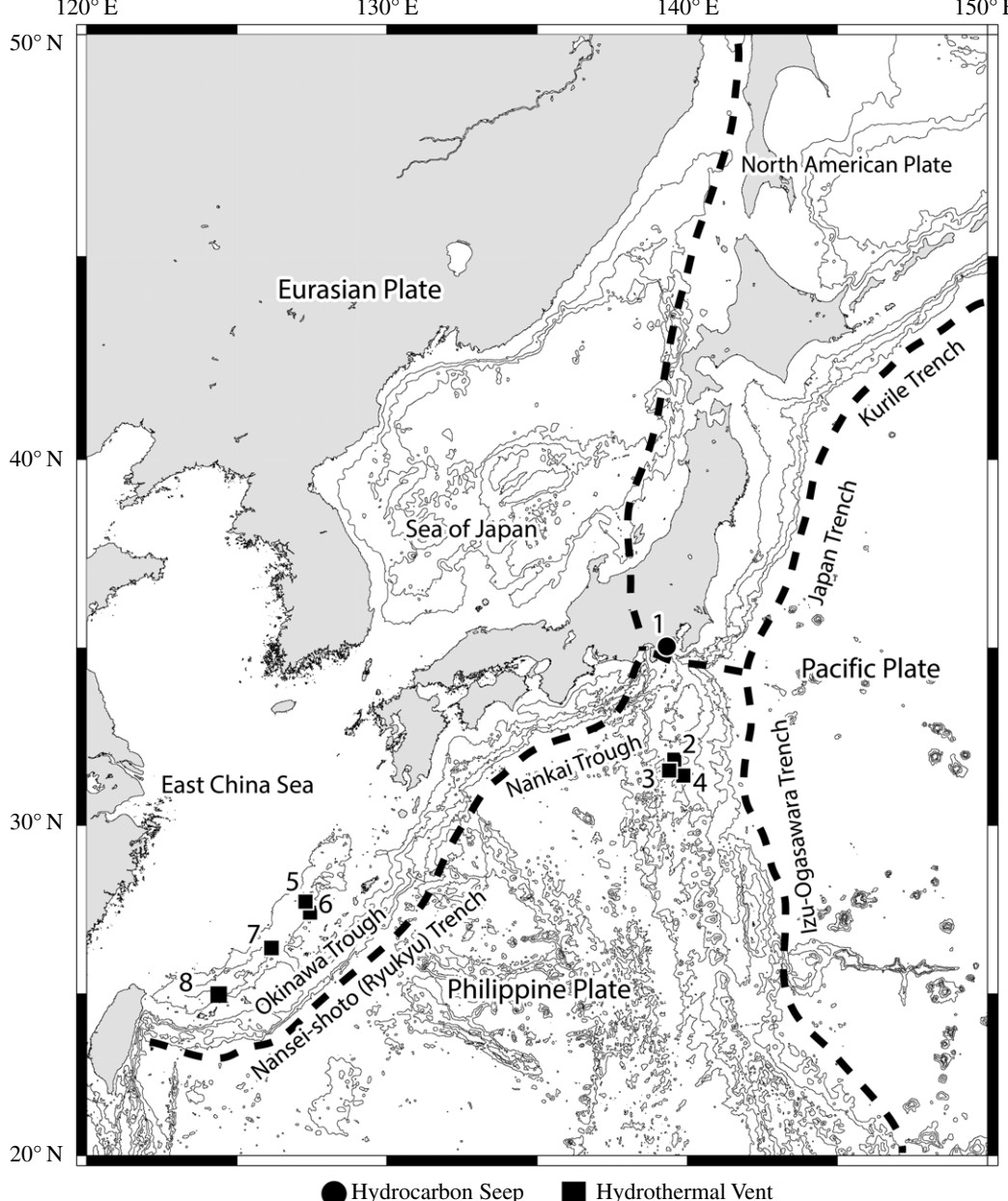

**Figure 1.** Map showing the location of Japanese chemosynthetic ecosystems relevant to the present study. 1, 'Off Hatsushima' site, Sagami Bay; 2, Myojin Knoll; 3, Bayonnaise Knoll Caldera; 4, Myojin-sho Caldera; 5, Sakai field; 6, Izena Hole; 7, Higa site, northeast of Gima Hill; 8, Crane site, Tarama Hill.

## 2.2. Morphology

The radula sac was dissected out under an Olympus SZX9 dissecting microscope. Any remaining soft tissue was cleaned using commercial bleach (concentration of sodium hypochlorite: 5%) diluted with four parts of water for 2–5 min until complete dissolution. The radula and operculum were mounted on metal stubs using carbon tape for scanning electron microscopy (SEM). SEM was carried out using a table top Hitachi TM3000 SEM (Japan Agency for Marine-Earth Science and Technology, JAMSTEC), for observation and imaging. The height and width of shells and their apertures were measured using digital Vernier callipers, rounded up to the nearest 1/10th millimetre value.

## 2.3. Genetics

Two specimens of each species were sequenced for the barcoding gene cytochrome oxidase $c$ subunit I (COI). Genomic DNA was extracted from a section of the foot in each specimen using QIAGEN

**Table 1.** Sampling localities.

| locality | area | habitat type | latitude | longitude | depth (m) | cruise |
|---|---|---|---|---|---|---|
| Bayonnase Knoll Caldera | Izu-Ogasawara | vent | 31°57.438′ N | 139°44.670′ E | 806 | NT07-17 |
| Myojin Knoll | Izu-Ogasawara | vent | 32°6.216′ N | 139°52.044′ E | 1244 | NT07-17 |
| Myojin-sho Caldera | Izu-Ogasawara | vent | 31°52.992′ N | 139°58.182′ E | 883 | NT07-17 |
| Hakurei site, Izena Hole | Okinawa Trough | vent | 27°14.815′ N | 127°04.089′ E | 1617 | NT11-20 |
| Higa/APA site, Gima Hill | Okinawa Trough | vent | 26°33.408′ N | 126°13.583′ E | 1483 | KR15-16 |
| Daisan-Kume Knoll | Okinawa Trough | vent | 26°18.356′ N | 126°24.855′ E | 1353 | KR16-04 |
| Crane Site, Tarama Hill | Okinawa Trough | vent | 25°4.539′ N | 124°31.010′ E | 1973 | KR15-16 |
| Sakai vent field | Okinawa Trough | vent | 27°31.013′ N | 126°58.960′ E | 1559 | KR15-17 |
| Off Hatsushima site | Sagami Bay | seep | 35°0.938′ N | 139°13.388′ E | 908 | KS16-04 |

DNeasy Blood and Tissue Kit following the manufacturer's instructions (QIAGEN, Tokyo, Japan), and quality checks of extractions were carried out using a Nanodrop 2000 spectrophotometer. The COI region was amplified with the primer pair LCO1490 and HCO2198 [21]. The polymerase chain reaction was carried out in 20 µl reactions, including 1 µl DNA template (15–30 ng µl$^{-1}$), 1 µl each of forward and reverse primers (10 µM), 1.6 µl dNTP mixture (TaKaRa Bio, Japan), 2 µl 10 × buffer, TaKaRa Ex Taq DNA polymerase solution (TaKaRa Bio, Japan) and 13.25 µl double-distilled water. A Veriti Thermal Cycler (Applied Biosystems) was used for thermo cycling. The protocol used was: 95°C for 1 min followed by 35 cycles of (95°C for 15 s, 40°C for 15 s, 72°C for 30 s), ending with 72°C for 7 min. Amplification was confirmed with 1.4% agarose gel electrophoresis using ethidium bromide. ExoSAP-IT (Affymetrix) was used following standard protocols to purify successful PCR products. Cycle sequencing reactions were carried out in 10 µl volumes, containing 1 µl PCR product, 0.5 µl BigDye Terminator v3.1 (Applied Biosystems), 0.7 µl 5× buffer, 0.25 µl primer (10 µM), 7.55 µl double-distilled water. The following protocol was used: 96°C for 1 min followed by 25 cycles of (96°C for 10 s, 50°C for 5 s, 60°C for 1 min 15 s), ending with 60°C for 4 min. Sequences were resolved from precipitated products using an Applied Biosystems 3130xl DNA sequencer (JAMSTEC).

The complimentary sequences from the forward and reverse primers were aligned to check the sequencing accuracy using the software package MEGA X [22] and Geneious R11 [23]. Phylogenetic reconstruction using Bayesian analysis was conducted using MrBayes 3.2 [24], with the most suitable nucleotide substitution model being selected by the Bayesian information criterion in PartitionFinder v. 2 [25]. The models selected were GTR + I + G for the first and the second codon and GTR + G for the third codon. In addition to new sequences obtained in the present study, abyssochrysoid COI sequences available on GenBank were used and sequences of the whelk *Neptunea amianta* (Dall, 1890), *Neptunea antiqua* (Linnaeus, 1758) and the periwinkle *Littorina littorea* (Linnaeus, 1758) from distantly related gastropod groups were included as outgroup taxa (after [16]). Metropolis-coupled Monte Carlo Markov chains were run for 2 million generations. Topologies were sampled every 100 generations, with the first 25% discarded as 'burn-in' to ensure chains convergence, which was confirmed using Tracer v.1.7 [26].

Restricted by the length of some shorter sequences on GenBank, the sequence length used in the final phylogenetic analyses was 470 bp. New sequences generated from this study are deposited in GenBank under the accession numbers MK560875–MK560877.

## 2.4. Specimen depository

Type specimens are deposited in collections of public museums, including the University Museum, the University of Tokyo (UMUT) and the Museum National d'Histoire Naturelle, Paris, France (MNHN).

# 3. Results

## 3.1. Systematics

Clade CAENOGASTROPODA Cox, 1960

 Superfamily ABYSSOCHRYSOIDEA Tomlin, 1927

 Family PROVANNIDAE Warén & Ponder, 1991

 Genus *Desbruyeresia* Warén & Bouchet, 1993

 **_Desbruyeresia armata_ n. sp.**

*ZooBank registration*: http://zoobank.org/urn:lsid:zoobank.org:act:63A20FDD-9124-404F-9B97-5C937C5965B8.

*Type material*: Holotype (figure 2*a–d*), UMUT RM33146. Paratypes: #1 (figure 2*g,h*), UMUT RM33147; #2 (figure 2*e,f*), MNHN-IM-2014-7112; #3 (figure 2*i,j*), MNHN-IM-2014-7113. All specimens live collected from the type locality, frozen in −20°C freezer and subsequently transferred to 99% ethanol. Dimensions are listed in table 2. As these are the only known specimens from the type locality, a piece of tissue from the foot of the holotype and paratype #3 were dissected for DNA sequencing. Paratype #3 was further dissected to obtain the radula and operculum for SEM imaging.

*Type locality*: Bayonnaise Knoll hydrothermal vent site, Izu-Ogasawara Arc, Japan; 31°57.438′ N, 139° 44.670′ E, 806 m deep; 2007/viii/29, ROV *Hyper-Dolphin* Dive #745, R/V *NATSUSHIMA* cruise NT07-17.

*Additional material examined:* Three specimens, live collected, frozen in −80°C. Myojin Knoll hydrothermal vent site, Izu-Ogasawara Arc, Japan; 32°6.216′ N, 139°52.044′ E, 1244 m; 2007/viii/26, ROV *Hyper-Dolphin* Dive #742, R/V *NATSUSHIMA* cruise NT07-17.

One specimen, live collected, fixed and stored in 70% ethanol. Myojin-sho Caldera hydrothermal vent site, Izu-Ogasawara Arc, Japan; 31°52.992′ N, 139°58.182′ E, 883 m deep; 2007/ix/01, ROV *Hyper-Dolphin* Dive #749, R/V *NATSUSHIMA* cruise NT07-17.

*Diagnosis*: A medium-sized *Desbruyeresia* up to 9.8 mm shell height, with inflated whorls and cancellate sculpture formed by strong axial ribbing and slightly weaker spiral ribbing. Tubercles or short spines at intersection between axial and spiral ribs. Radula teeth thin and membranous with weak support. Marginal teeth only coarsely serrated into four denticles.

*Description*: Shell (figure 2) broad melanoid in form, thick, solid. Protoconch unknown due to all specimens having corroded apex. Most specimens heavily corroded with only three to four whorls remaining. Whorls inflated and rapidly expanding for its genus, bearing rough, cancellate sculpture. The posterior of periphery with 12–14 axial ribs and three strong spiral ribs, usually increasing to four to five on body whorl. Tubercles formed at intersection of axial and spiral ribs, often developing into short spines. Suture distinct, well defined. Basal area anterior of periphery marked by three to four weaker spiral ribs, axial sculpture obsolete. Aperture oval, taller than wide. Columella curved anteriorly. Only a very shallow siphonal fasciole present.

Periostracum thick, bright yellowish green in coloration. Most specimens further covered by a thick, black coating of sulfide deposit.

Corneous operculum (figure 3*b*) typical of genus present, membranous and semi-transparent. Yellowish-brown in coloration, about three whorls. Pausispiral, teardrop-shaped, bluntly pointed. Nucleus eccentric.

Radula (figure 3*a*) taenioglossate, formula $2 + 1 + 1 + 1 + 2$. Teeth membranous, flat and broad overall. Central tooth triangular. Main cusp large, moderately sharp, with three very small denticles on either side. Central and lateral supporting ridges weak, poorly developed. Lateral teeth with a large, spade-like central cusp. Inner cutting edge of lateral carry one to three conspicuous denticles, outer cutting edge with four to five minor denticles. Shafts of laterals carry a blunt horizontal ridge below apical part. Marginals flat, very broad. Apical part carries four large cusps, outermost cusp strongest with sigmoidal inner edge, other three cusps decrease in strength towards rachidian. No other minor denticles present.

Soft parts were unavailable for examination as they were fixed for molecular work.

*Distribution*: Known from three hydrothermal sites in the Izu-Ogasawara Arc, including Bayonnaise Knoll, Myojin Knoll, and Myojin-sho Caldera.

*Etymology*: 'Armatus' (Latin), meaning armed or armoured. This refers to the shell being armed with short spines and tubercles as well as 'armoured' with a thick layer of sulfide. Used as an adjective.

*Remarks*: The radula characteristics of the present new species prevent it being confused with any other described *Desbruyeresia* species. Particularly, the broad, flat marginal teeth with the distal end only coarsely serrated into four denticles is clearly distinct from most other *Desbruyeresia* which have the distal end finely serrated into 10 or more denticles [9,15,27]. Only *D. marisindica*, Okutani

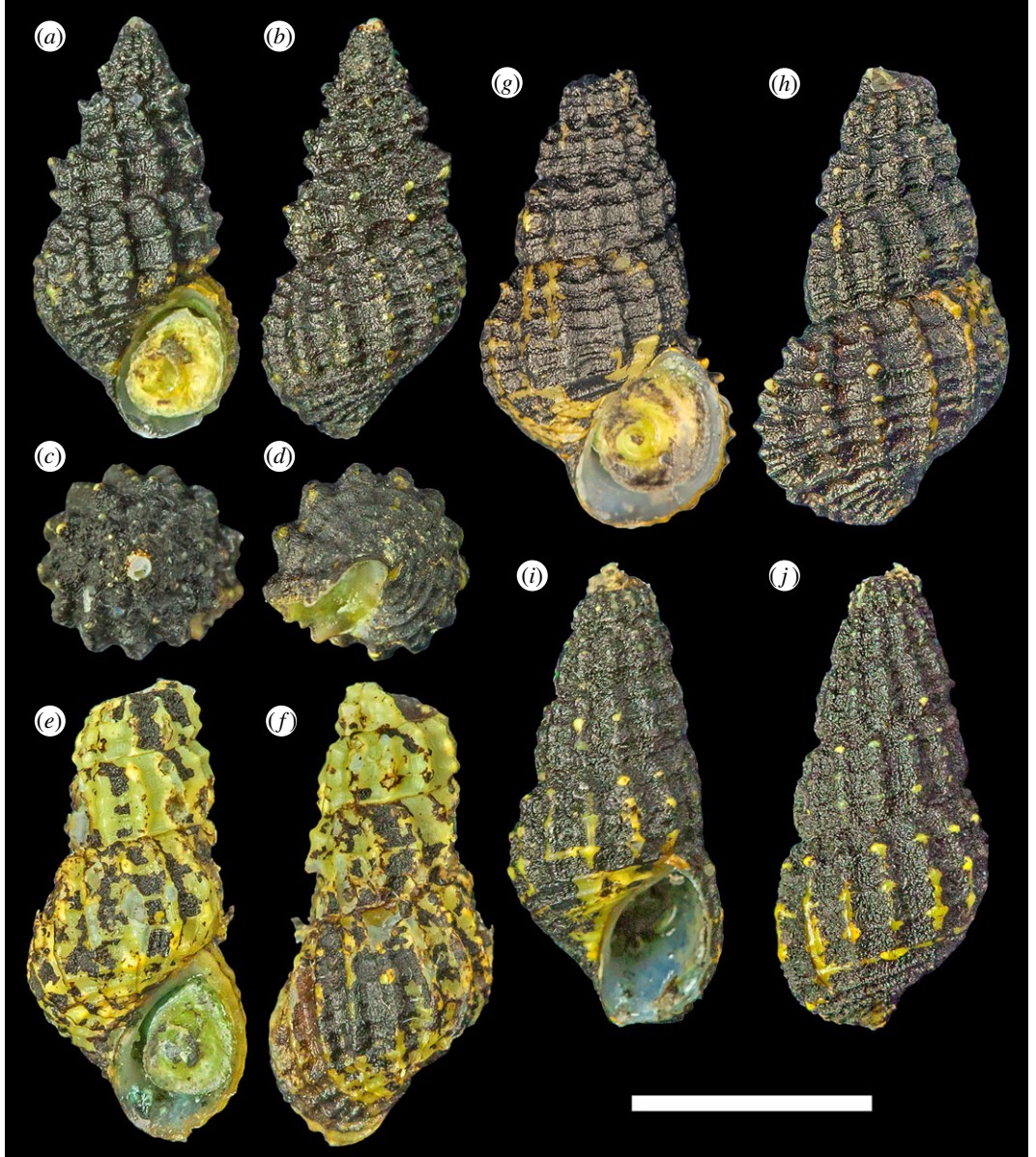

**Figure 2.** *Desbruyeresia armata* n. sp. (*a–d*) Holotype, shell height 8.8 mm (UMUT RM33146). (*e,f*) Paratype #2, shell height 9.8 mm (MNHN-IM-2014-7112). (*g–h*) Paratype #1, shell height 9.8 mm (UMUT RM33147). (*i,j*) Paratype #3, shell height 9.8 mm (MNHN-IM-2014-7113). Scale bar, 5 mm, applies to the entire figure.

Hashimoto & Sasaki, 2004, has similar marginal teeth, but even in this species, the apical part is serrated into six to eight denticles [28]. The central cusp of the rachidian is also much larger in *D. armata* n. sp. compared to that of *D. marisindica* [28]. The external morphology of the present new species is most similar to *Desbruyeresia marianensis* from Mariana back-arc basin [27], from which it can be distinguished by the coarser sculpture and the tendency to form short, blunt spines in the intersection between axial and spiral ribs, a feature lacking in *D. marianensis*. *Desbruyeresia armata* n. sp. also has fewer (3–4) spiral ridges on the base compared to *D. marianensis* (5; [27]). The new species can also be compared with *Desbruyeresia spinosa* Warén & Bouchet, 1993, from North Fiji and Lau basins, the other *Desbruyeresia* species known to form spines. The spines of *D. spinosa*, however, are much longer and sharper than in the present new species [9]. Furthermore, the whorls of *D. spinosa* appear much more angular than *D. armata* n. sp. due to the longer spines as well as a deeper suture [9].

### *Desbruyeresia costata* n. sp.

*ZooBank registration:* http://zoobank.org/urn:lsid:zoobank.org:act:A2ED2520-F4C4-4B30-9688-1FC71668B371.

**Table 2.** Shell dimensions of the type specimens.

| species | type specimen | shell height (mm) | shell width (mm) | aperture height (mm) | aperture width (mm) |
|---|---|---|---|---|---|
| *Desbruyeresia armata* n. sp. | holotype | 8.8 | 4.3 | 3.3 | 2.2 |
| | paratype 1 | 9.7 | 5.3 | 4.2 | 3.2 |
| | paratype 2 | 9.8 | 4.7 | 4.2 | 2.5 |
| | paratype 3 | 9.8 | 4.9 | 4.2 | 2.7 |
| *Desbruyeresia costata* n. sp. | holotype | 9.3 | 4.3 | 3.6 | 2.3 |
| | paratype 1 | 9.0 | 3.8 | 2.9 | 2.0 |
| | paratype 2 | 9.1 | 3.9 | 3.7 | 2.3 |
| | paratype 3 | 9.0 | 4.7 | 3.6 | 2.6 |
| | paratype 4 | 10.5 | 5.7 | 4.4 | 2.8 |
| *Provanna fenestrata* n. sp. | holotype | 7.1 | 5.2 | 3.9 | 2.6 |
| | paratype 1 | 7.2 | 5.4 | 4.1 | 2.7 |
| | paratype 2 | 6.5 | 4.8 | 3.7 | 2.5 |
| | paratype 3 | 6.8 | 4.5 | 3.4 | 2.3 |
| | paratype 4 | 6.7 | 4.6 | 3.7 | 2.6 |
| *Provanna stephanos* n. sp. | holotype | 7.1 | 5.5 | 4.3 | 2.8 |
| | paratype 1 | 5.1 | 4.5 | 3.0 | 2.4 |

*Type material*: Holotype (figure 4*a–d*), UMUT RM33148. Paratypes: #1 (figure 4*e,f*), one specimen, MNHN-IM-2014-7114; #2 (figure 4*g,h*), one specimen, MNHN-IM-2014-7115; #3 (figure 4*i,j*), one specimen, UMUT RM33149; #4, one specimen, MNHN-IM-2014-7116. All type specimens were collected from the type locality, fixed and stored in 99% ethanol. Paratype #4 was fresh dead collected without soft parts or operculum, all other type specimens were live collected. Dimensions are listed in table 2.

*Type locality*: Hakurei site, Izena Hole hydrothermal vent field, Okinawa Trough, Japan; 27°14.815′ N, 127°04.089′ E, 1617 m deep, 2011/x/05, ROV *Hyper-Dolphin* Dive #1329, R/V *NATSUSHIMA* cruise NT11-20.

*Additional material examined*: Three further live-collected specimens from the type locality, shell decalcified for molecular work and radula observations.

One live-collected specimen and two dead shells. Higa hydrothermal vent field, northeast of Gima Hill, Okinawa Trough, Japan; 26°33.408′ N, 126°13.583′ E, 1483 m deep; ROV *KAIKO* (with vehicle *Mk-IV*) Dive #671, R/V *KAIREI* cruise KR15-16, 2015/xi/02.

*Diagnosis*: A medium-sized *Desbruyresia* up to 10.5 mm shell height, with 10–12 strong axial ribs on the body whorl. Spiral ribs, other than the two to three present on the base, are present but weak and indistinct except on early whorls. Radula typical of *Debruyeresia*.

*Description*. Shell (figure 4) melanoid in form, rather thick, solid. Protoconch unknown due to all specimens having corroded apex. Only three to four whorls remain on most specimens. Whorls highly elevated, slowly expanding. The posterior of periphery marked by strong axial ribs numbering 10–12, three spiral ribs present but only distinct on early whorls, becoming obsolete on adult shells. Suture distinct, impressed in aged individuals. The interior of periphery axial ribs becomes obsolete, instead three weak but conspicuous spiral ribs appear. The aperture oval in shape, taller than wide. Columella variable, either straight or becoming moderately curved. Siphonal notch lacking, only a shallow fasciole present. Growth lines indistinct except the body whorl, where some clear lines may be present.

Periostracum thick, yellowish-brown in coloration, sometimes carry a hint of red. Some sulfide deposit present on most specimens but coverage not extensive.

Corneous operculum present, membranous and semi-transparent (figure 3*d*). Yellowish-brown in coloration. Teardrop-shaped, bluntly pointed at top. Pausispiral, nucleus eccentric.

Radula (figure 3*c*) taenioglossate with formula $2 + 1 + 1 + 1 + 2$. Teeth solid. Central tooth triangular, with sharp-tipped main cusp, two to three strong denticles on either side. Central and lateral supporting

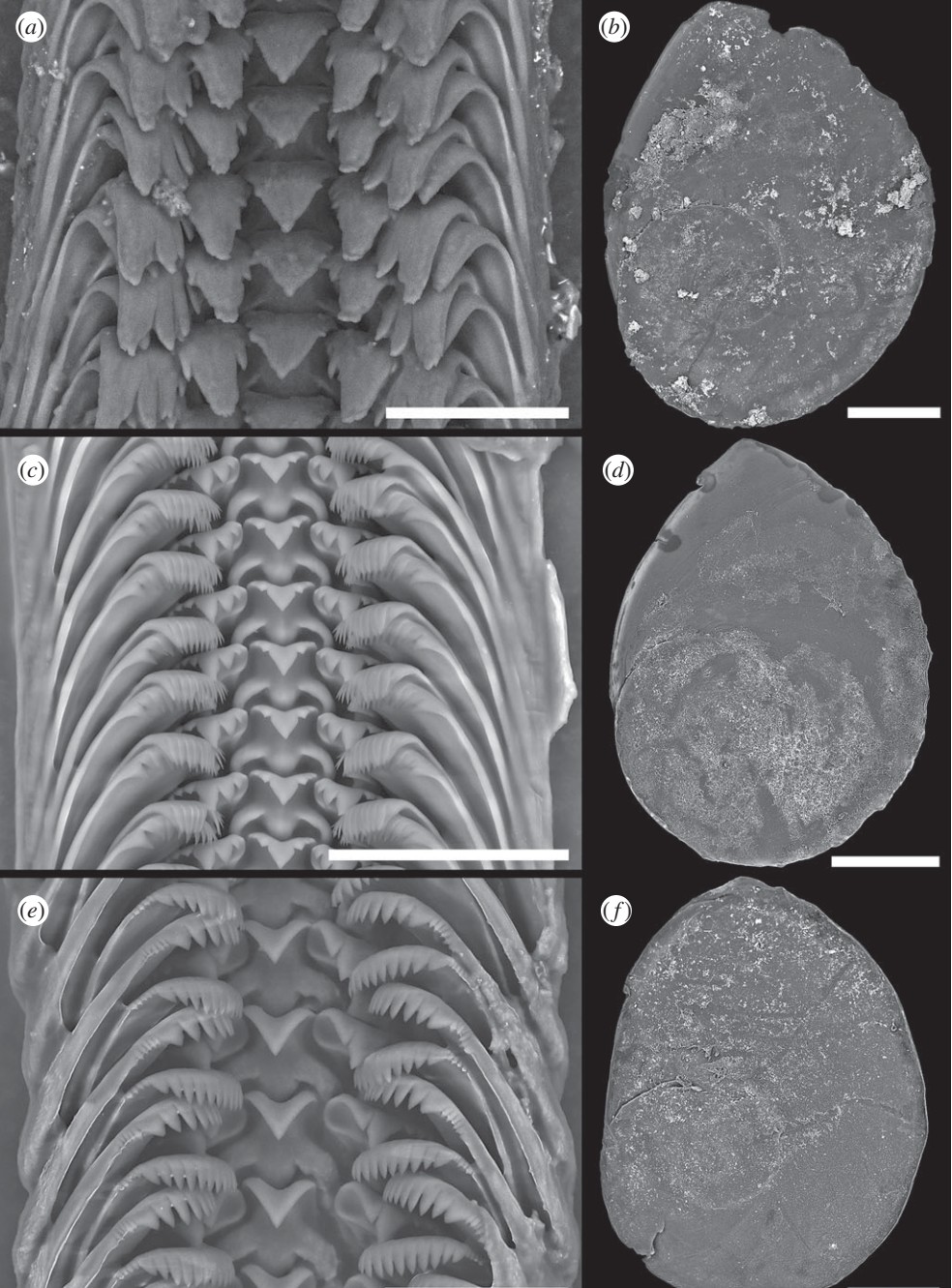

**Figure 3.** Scanning electron micrographs. (*a,b*) *Desbruyeresia armata* n. sp.: (*a*) radula, (*b*) operculum; (*c,d*) *Desbruyeresia costata* n. sp.: (*c*) radula, (*d*) operculum; (*e*–*f*) *Provanna fenestrata* n. sp.: (*e*) radula, (*f*) operculum. Scale bars: *a,c,e*, 50 µm; *b,d,f*, 500 µm.

ridges strong, well developed. Lateral teeth with elongate, well-developed, finely pointed central cusp, inner edge sigmoidal in outline. Inner cutting edge of laterals equipped with three denticles, outer cutting edge with four to five much smaller denticles. One well-developed, sharp protrusion present just below apical part. Marginals flat, broad, distally truncated, apical part rake-like and evenly serrated into *ca* 12–14 denticles. No other minor denticles present. Serration slightly finer in outer marginals than inner marginals.

Soft parts were fixed for molecular work and consequently were unsuitable for anatomical investigation.

*Distribution*: Hitherto only known from the two hydrothermal vent fields in the mid-Okinawa Trough: Izena Hole and Higa site.

*Etymology*: '*Costatus*' (Latin), meaning ribbed. This refers to the strong and conspicuous axial ribs in this species. Used as an adjective.

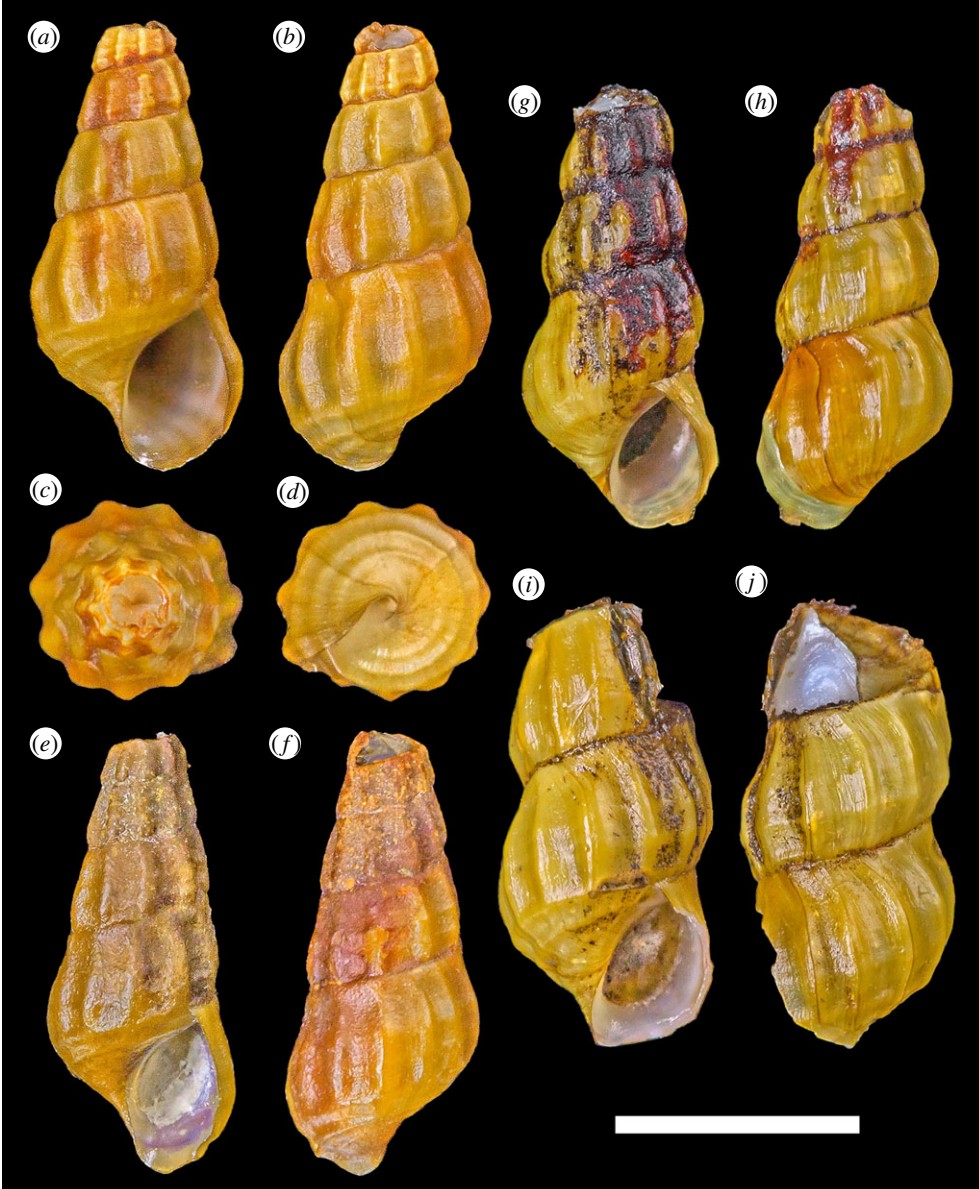

**Figure 4.** *Desbruyeresia costata* n. sp. (*a*–*d*) Holotype, shell height 9.3 mm (UMUT RM33148). (*e,f*) Paratype #1, shell height 9.0 mm (MNHN-IM-2014-7114). (*g,h*) Paratype #2, shell height 9.1 mm (MNHN-IM-2014-7115). (*i,j*) Paratype #3, shell height 9 mm (UMUT RM33149); *j* is a side view to show the calcified 'plug' inside the corroded apex. Scale bar, 5 mm, applies to the entire figure.

*Remarks*: *Desbruyeresia costata* n. sp. is distinct from other described *Desbruyeresia* species from its shell sculpture, in which the spiral ribs are obsolete in the adult whorls. In the early whorls of *D. costata* n. sp., the spiral ribs are stronger (figure 4*a,b*) but still much weaker than the axials. In all other reported *Desbruyeresia* species, the spiral sculpture remains strong throughout growth stages, and the shell surface always bears a latticed sculpture from crossings of the axial and spiral ribs [9,15,27,28]. The spire is often very heavily corroded away in large specimens (figure 4*i*), the opening being 'plugged' by a thick layer of secondary shell formation (figure 4*j*).

Genus *Provanna* Dall, 1918

**Provanna fenestrata n. sp.**

*ZooBank registration*: http://zoobank.org/urn:lsid:zoobank.org:act:BA91C2C1-6BBC-409C-9A33-BE4DFEA9A53C.

*Type material*: Holotype (figure 5*a*–*d*), UMUT RM33150. Paratypes: #1 (figure 5*e,f*), MNHN-IM-2014-7117; #2 (figure 5*g,h*), MNHN-IM-2014-7118; #3 (figure 5*i,j*), UMUT RM33151; #4 (figure 5*k,l*),

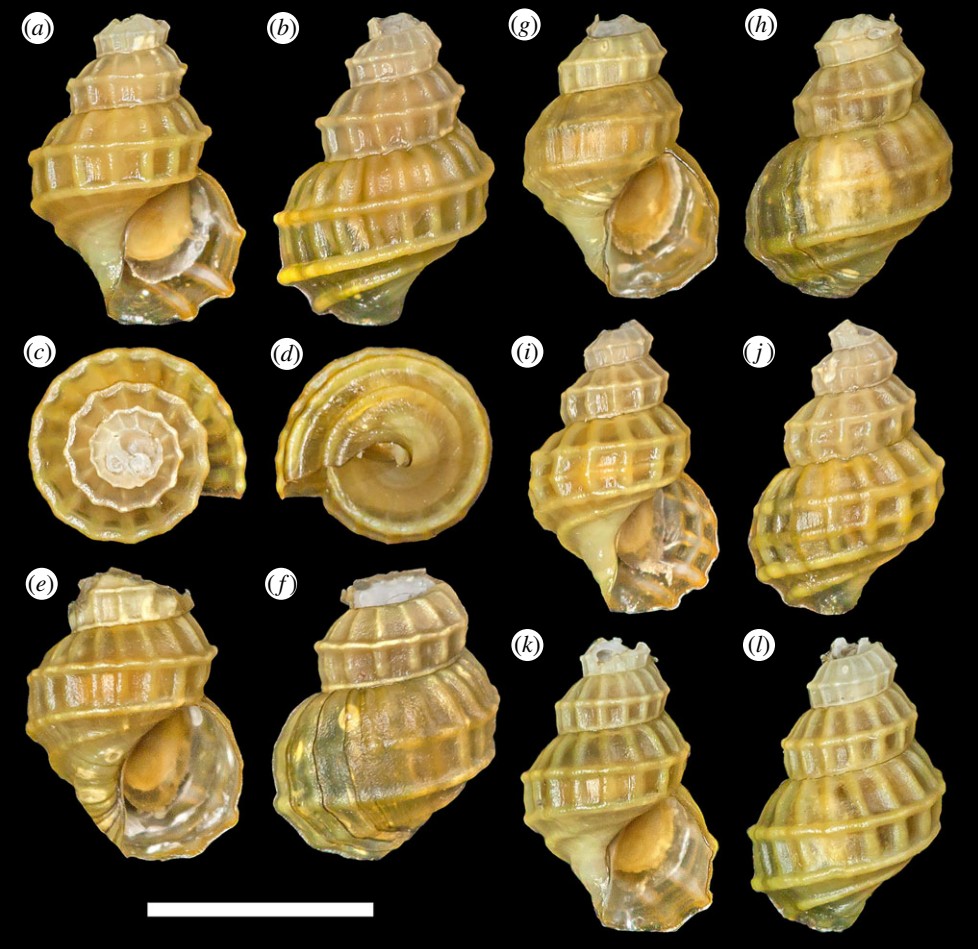

**Figure 5.** *Provanna fenestrata* n. sp. (*a–d*) Holotype, shell height 7.1 mm (UMUT RM33150). (*e,f*) Paratype #1, shell height 7.2 mm (MNHN-IM-2014-7117). (*g,h*) Paratype #2, shell height 6.5 mm (MNHN-IM-2014-7118). (*i,j*) Paratype #3, shell height 6.8 mm (UMUT RM33151). (*k,l*) Paratype #4, shell height 6.7 mm (MNHN-IM-2014-7119). Scale bar, 5 mm, applies to the entire figure.

MNHN-IM-2014-7119. Holotype and paratypes are fixed and stored in 99% ethanol. All type specimens were live collected from the type locality. Dimensions are listed in table 2.

*Type locality*: Crane site, Tarama Hill, Okinawa Trough, Japan [29]; 25°4.539′ N, 124°31.010′ E, 1973 m deep; 2015/x/29, ROV *KAIKO* (with vehicle *Mk-IV*) Dive #669, R/V *KAIREI* cruise KR15-16.

*Additional material examined*: Two live-collected specimens, fixed and stored in 99% ethanol, from the type locality. Used for DNA sequencing and SEM imaging of the radula and operculum.

Two live-collected specimens, fixed and stored in 99% ethanol. Sakai hydrothermal vent field [20], 27°31.013′ N, 126°58.960′ E, 1559 m deep; 2015/xi/13, ROV *KAIKO* (with vehicle *Mk-IV*) Dive #676, R/V *KAIREI* cruise KR15-17.

*Diagnosis*. A medium-sized *Provanna* up to 7.2 mm shell height, characterized by strong and raised, equally spaced axial and radial ribs intersecting to form regular rectangular lattice sculpture.

*Description*. Shell (figure 5) thin, translucent. Soft parts visible through shell. Ostracum often partly or completely corroded in body whorl leaving only periostracum layer. Protoconch unknown due to all specimens having corroded apex. Whorls highly elevated, inflated for its genus. Sculpture posterior of periphery consists of two (rarely three) spiral ribs and 16–20 axial ribs approximately equal in strength and spacing. Spiral and axial ribs intersect to form remarkably methodical sculpture of regular rectangles in a lattice-fashion. Most anterior spiral rib just above or at the same position as suture. Suture distinct, well defined, slightly impressed. The anterior of periphery marked by one rather strong spiral rib and further one or two much weaker ones further anterior to it. Axial ribs become obsolete at this point. Aperture semicircular, slightly taller than wide. Columella very variable in form, ranging from straight to sigmoidal to curved to one side. Shallow but distinct siphonal notch present. Growth line indistinct in early whorls, body whorl often marked by some conspicuous growth lines.

Periostracum thick, yellowish green in coloration.

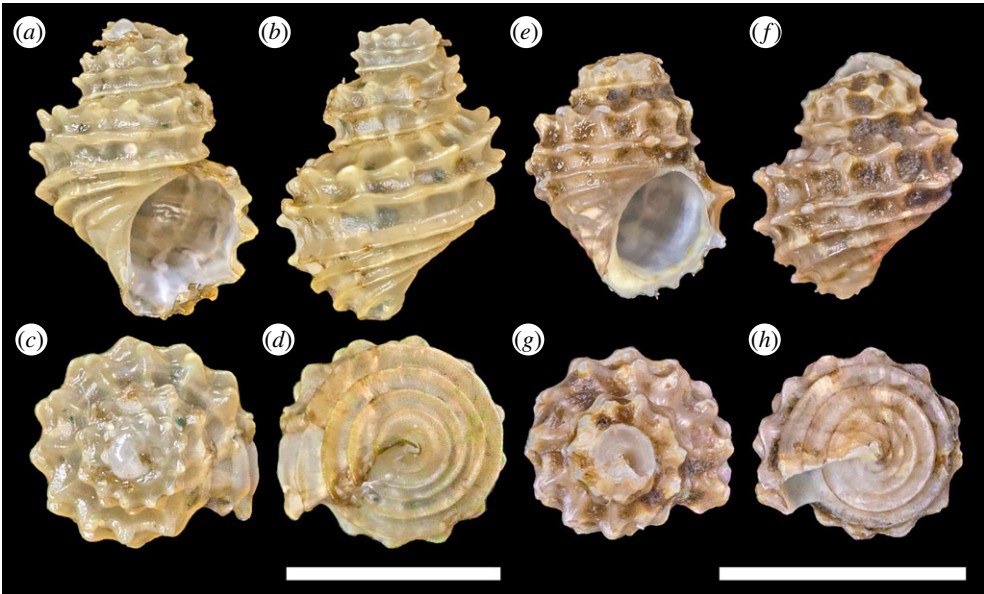

**Figure 6.** *Provanna stephanos* n. sp. (*a–d*) Holotype, shell height 7.1 mm (UMUT RM33152). (*e–h*) Paratype #1, shell height 5.1 mm (MNHN-IM-2014-7120). Both scale bars, 5 mm.

Oval shaped, bluntly pointed operculum present (figure 3*f*). Pausispiral with eccentric nucleus, a little more than 3.5 whorls. Thin, semi-transparent, yellowish-brown in coloration.

Radula (figure 3*e*) taenioglossate, formula 2 + 1 + 1 + 1 + 2. Teeth solid. Central tooth with a single triangular, sharply pointed main cusp. Central and lateral supporting ridges well developed. Lateral teeth with a fine-tipped main cusp, inner edge sigmoidal in outline. A single further cusp present inside of the main cusp. Three to four minor denticles present outside the main cusp on the outer cutting edge. Below apical part of laterals a single sharp protrusion is present on shaft. Marginal teeth rather flat, broad, distally truncated. Apical part rake-like, evenly serrated into *ca* 12–14 denticles. The outermost part of the cutting edge carries a further four to six minor denticles. Serration slightly finer in inner than outer marginals.

Gross external anatomy of soft parts was examined and found to match a previous description of the genus by Waren & Ponder [30] very well. Entire animal occupies 1.7–2.2 whorls. Foot large, lacking epipodial tentacles. Head with flattened snout and one pair of thick cephalic tentacles. Pallial edge smooth. Gonads situated on the apex of visceral mass, followed by the digestive gland and a sizeable stomach. The kidney situated below the digestive gland, to the right side of the ventricle. A single gill occupies about 0.7 whorl. The hypobranchial gland between the gill on the left and the rectal sinus on the right when viewed from exterior. The pallial part of the gonad situated posterior of the rectal sinus.

*Distribution*: So far only known from two hydrothermal vent sites in the mid-Okinawa Trough: Crane site, Tarama Hill and the Sakai vent field.

*Etymology*: 'Fenestrata' (Latin, feminine) meaning fenestrated or furnished with windows, referring to the numerous characteristically regular and consistent rectangular grid-like lattice sculptures made by intersecting radial and axial ribs. Used as an adjective.

*Remarks*: The present new species is highly unique in its sculpture and we are unaware of any described *Provanna* species that can be confused with it. It resembles *P. segonzaci* Warén & Ponder, 1991 from Lau Basin [30,31] and *P. clathrata* Sasaki *et al.*, 2016 from Okinawa Trough [7] to a certain extent, but can be distinguished from these two species by its coarser, more regular lattice sculpture.

### *Provanna stephanos* n. sp.

*ZooBank registration*: http://zoobank.org/urn:lsid:zoobank.org:act:9A2C775A-606F-4BBA-9B25-956F1BB220A4.

*Type material*: Holotype (figure 6*a–d*), UMUT RM33152. Paratype #1 (figure 6*e–h*), MNHN-IM-2014-7120. Both type specimens were fresh dead collected without soft parts or operculum, stored in 99% ethanol to prevent desiccation of periostracum. Only known from the type series. Dimensions are listed in table 2.

*Type locality*: 'Off Hatsushima' seep site, Sagami Bay, central Honshu, Japan; 35°0.938′ N, 139°13.388′ E, 908 m deep; 2016/v/08, ROV *Hyper-Dolphin* Dive #1965, R/V *SHINSEI MARU* cruise KS16-04.

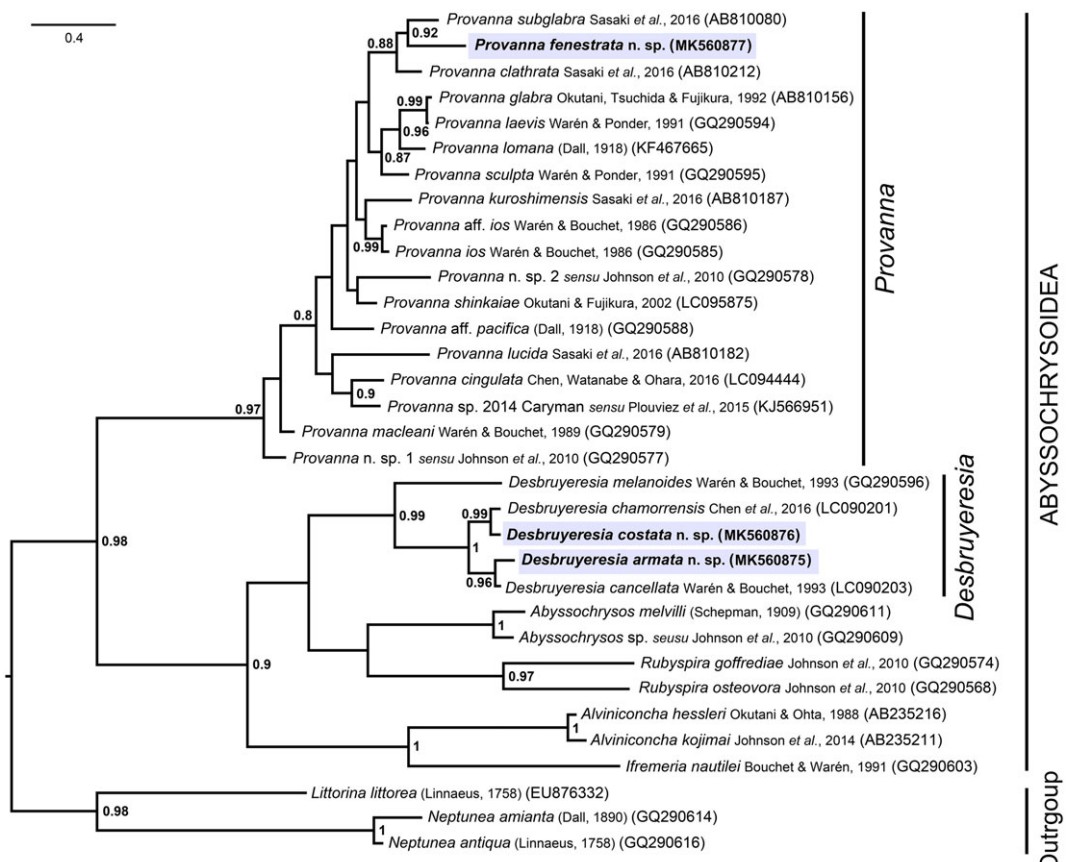

**Figure 7.** Phylogenetic tree of Abyssochrysoidea reconstructed by Bayesian inference using 470 bp of the COI gene, showing the systematic position of the three new species for which genetic data were available. Node values represent Bayesian posterior probabilities, only shown when over 0.8.

*Diagnosis*. A medium-sized *Provanna* up to 7.1 mm shell height with very strong spiral ribbing, 3 above suture and 4 below. The two most posterior spiral ribs form characteristic sturdy, short spines upon intersection with weaker axial ribs. The periostracum is light olive and not yellowish as in many other species of *Provanna*.

*Description*. Shell thin, translucent. The apex corroded with only about three whorls remaining, protoconch therefore unknown. Whorls highly elevated, rather inflated. Suture distinct. Sculpture above periphery consist of three strong, raised spiral ribs and 12 consistently spaced axial ribs. Axial ribs strong at posterior half of a whorl but disappear before reaching periphery. Two most posterior spiral ribs form conspicuous, sturdy, short spines at intersection with axial ribs, while the anterior one remains smooth due to axial ribs disappearing. Four further roughly equally spaced spiral ribs present in the basal area anterior of periphery, increasing in strength posteriorly. Aperture semicircular, slightly taller than wide. Columella straight. An indistinct siphonnal notch present. Growth line indistinct.

Periostracum rather thick, light olive in coloration.

*Distribution*: Only known from the type locality.

*Etymology*: 'Stéphanos' (Greek), literally 'that which encircles or surrounds' and meaning crown or wreath. This refers to the two spinous spiral cords which resemble a laurel wreath or a crown. Used as a noun in apposition.

*Remarks*: This new species with a remarkably spinous sculpture was collected together with large holobiont bivalves *Bathymodiolus japonica* Hashimoto & Okutani, 1994, *B. platifrons* Hashimoto & Okutani, 1994, and *Phreagena okutanii* (Kojima & Ohta, 1997). Although only two fresh dead specimens without soft parts or opercula were collected, its characteristic sculpture with two rows of strongly spinous spiral ribs is unique and highly distinct among all known *Provanna* or *Desbruyeresia* species and warrants its position as a new species. Although *Provanna goniata* Warén & Bouchet, 1986, from the Gulf of California, has a somewhat similar sculpture, but the spines in that species are much weaker than the present new species [32]. Given that information regarding the protoconch, radula and DNA is lacking for the present new species, it is difficult to assign it to either *Provanna* or

*Desbruyeresia* with absolute certainty. It is here assigned to *Provanna* on the basis of its shell which is short and stout in proportion with a distinct siphonal notch, more typical of *Provanna*.

## 3.2. Molecular phylogeny

Molecular phylogeny reconstruction of Abyssochrysoidea carried out with Bayesian inference using 470 bp of the COI gene (figure 7) recovered the superfamily as a monophyletic clade. All currently recognized abyssochrysoid genera that were included in the phylogeny were recovered as monophyletic clades, including *Provanna* (Bayesian posterior probability, BPP = 0.97) and *Desbruyeresia* (BPP = 0.99), which were both strongly supported. The generic assignment of the three new species for which molecular data were available, namely *D. armata* n. sp., *D. costata* n. sp., and *P. fenestrata* n. sp., were confirmed by the phylogeny.

Within *Desbruyeresia*, *D. armata* n. sp. was found to be closest to *D. cancellata*, Warén & Bouchet, 1993, among the species included (BPP = 0.96), with an uncorrected *P*-distance between the two being 6.2% over 470 bp. The two can be distinguished morphologically by *D. armata* n. sp. having only four denticles on the marginal teeth, with *D. cancellata* having 12, and by the fact that short spines are never formed on *D. cancellata* [9]. *Desbruyeresia costata* n. sp. was shown to be closely related to *D. chamorrensis* [15] (BPP = 0.99), separated by uncorrected *P*-distance of 5.9% (470 bp, COI gene). *Desbruyeresia costata* n. sp. and *D. chamorrensis* and are morphologically distinct, as the shell sculpture of *D. costata* lacks strong spiral cords that are highly conspicuous on *D. chamorrensis* [15]. These two pairs of sister taxa were in turn sister to each other, forming a fully supported clade within *Desbruyeresia* (BPP = 1).

Within *Provanna*, *P. fenestrata* n. sp. was recovered sister to *P. subglabra* Sasaki *et al.*, 2016 (BPP = 0.92) with an uncorrected *P*-distance of 7.0% between the two; they are easily separable morphologically since *P. subglabra* has a smooth shell surface lacking in significant surface sculpture, very different from the strong, raised cancellate sculpture seen in *P. fenestrata* n. sp. The pair was in turn sister to *P. clathrata*, another species endemic to the Okinawa Trough, with a moderate support (BPP = 0.88).

# 4. Discussion

## 4.1. Morphological differences between *Desbruyeresia* and *Provanna*

Originally, the two abyssochrysoid genera, *Desbruyeresia* and *Provanna*, were separated by the absence of a pallial tentacle in *Desbruyeresia*, but this was later confirmed to be present in better preserved specimens [9,33]. The protoconches of the two genera are drastically different, however. Most *Desbruyeresia* have a tall, multispiral protoconch with cancellate sculpture and both protoconch I and II are present indicating planktotrophic dispersal (except *D. marisindica* which is lecithotrophic); by contrast, *Provanna* has a protoconch typical of lecithotrophic dispersal consisting of 1.5 whorls and lacking in protoconch II, sculptured mainly by axial ribs [6,9,11,30,33]. In adult specimens, the apex is generally corroded off and the protoconch is usually unavailable for observation.

Generally speaking, *Desbruyeresia* has a taller, more slender shell than *Provanna*, but this is not conclusive on its own. Another teleoconch feature that is rather useful is that *Provanna* tends to have a distinct siphonal notch, which is reduced to a very shallow fasciole in *Desbruyeresia*.

We note that the most realistic way to distinguish the two genera is from two aspects of radula characteristics. Firstly, the central tooth of *Desbruyeresia* carries lateral cusps in addition to the main cusp; these lateral cusps are lacking in *Provanna* [11]. Secondly, the outermost cutting edge of the marginal teeth is smooth in *Desbruyeresia*, but a series of minor, fine cusps are present in *Provanna* (perhaps except *P. pacifica* which has a distinctive radula, see fig. 17 in [32]).

The marginal tooth of *D. armata* n. sp. is highly distinct in that it is only divided into four broad denticles instead of finely serrated as is the norm for the genus. Given its phylogenetic position (figure 4) well nested within *Desbruyeresia*, this is most likely a recently acquired character, perhaps linked with the substrate type upon which it grazes upon. For other species described herein, the radula was not significantly different from their congeners in the Western Pacific and was of little value for species delineation, as already pointed out by Sasaki *et al.* [7].

## 4.2. Biogeography

By analysing the population genetics of five *Provanna* species inhabiting chemosynthetic ecosystems around Japan, including *P. glabra*, *P. subglabra*, *P. lucida*, *P. kuroshimensis* and *P. clathrata* across

chemosynthetic ecosystems ranging between 644 and 1646 m in depth, Ogura *et al.* [34] showed that these species were segregated by depth. *Provanna fenestrata* n. sp. reported herein appears to be a deep-water specialist occurring between depths of 1559 and 1973 m, which is even deeper than *P. clathrata*, the deepest occurring species reported in Ogura *et al.* [34]. This further reinforces the idea that water depth may be a key factor in their separation and evolution [34]. Although *Provanna* species often occur in high-density aggregates, the surprising discovery of *Provanna stephanos* n. sp. from the very well surveyed off Hatsushima seep site [4] indicates that further, rare lineages of *Provanna* are likely to be present in sites largely dominated by other congeners, in this case *P. glabra*. This also indicates that the total number of specimens, records, and discovered localities of the genus is still comparatively small, which makes a comprehensive understanding of its zoogeography difficult.

The two newly described *Desbruyeresia* species also occur at different depths, with *D. armata* n. sp. occurring between 806 and 1244 m and *D. costata* between 1483 and 1617 m; but given that *D. armata* n. sp. is only known from the Izu-Ogasawara Arc and *D. costata* n. sp. only from the Okinawa Trough, the two species are likely separated by geographical barriers between the two regions. The fact that the sister taxa recovered on the phylogenetic reconstruction for both *D. armata* n. sp. and *D. costata* n. sp. occur more southerly in the western Pacific area [11] suggests that the two species represent separate invasions of the genus into the northern part of the western Pacific. Including these new species, there is currently no record of abyssochrysoids occurring across the Okinawa Trough and Izu-Ogasawara Arc, or species occurring in both vents and seeps in the western Pacific [7].

Ethics. The faunal collections were conducted in Japanese exclusive economic zone by Japanese government research vessels. All applicable international, national, and/or institutional guidelines for the care and use of animals were followed by the authors. Research animals were invertebrate gastropod molluscs and no experiments with animals were conducted in this study.

Data accessibility. DNA sequences: GenBank accessions MK560875–MK560877.

Authors' contributions. C.C. and T.S. conceived and designed the project. C.C. and H.K.W. collected and fixed the samples. C.C. dissected the samples. H.K.W. extracted DNA and performed sequencing; C.C. and H.K.W. carried out genetic and phylogenetic analyses. C.C. interpreted the data and drafted the manuscript. All authors contributed to the final manuscript.

Competing interests. The authors declare no competing interests.

Funding. The research cruises YK15-14, KR15-16 and KR16-04 were supported by Council for Science, Technology, and Innovation (CSTI) as the Cross Ministerial Strategic Innovation Promotion Program (SIP), Next-generation Technology for Ocean Resource Exploration. The genetic work was funded by two Japan Society for the Promotion of Science KAKENHI grants, one (15K18602) awarded to H.K.W. and another (18K06401) to H.K.W. and C.C. The funders had no role in study design, data collection and analysis, decision to publish or preparation of the manuscript.

Acknowledgements. The authors would like to express their gratitude to the principal scientists of the relevant scientific expeditions: Kokichi Iizasa (NT07-17), Jun-ichiro Ishibashi (NT11-20), Akinori Yabuki (KS16-04), Shinsuke Kawagucci (KR15-16), Kentaro Nakamura (YK15-14), and Hiroyuki Yamamoto (KR15-17). Captains and crews of the JAMSTEC R/Vs *KAIREI* (KR15-16, KR15-17), *NATSUSHIMA* (NT07-17, NT11-20), *YOKOSUKA* (YK15-14), and *SHINSEI MARU* (KS16-04), as well as the ROVs *KAIKO* (with vehicle *Mk-IV*) and *Hyper-Dolphin* teams on the same expeditions are thanked for their tireless support of the scientific activity during the cruises. Ken Nagata and Taiga Kijima kindly helped in obtaining the new molecular sequences used in this work. We thank Shinsuke Kawagucci for useful comments and insightful discussions.

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
