## [Reviewer comments · Royal Society Open Science]

Review History

RSOS-190393.R0 (Original submission)

Review form: Reviewer 1

Is the manuscript scientifically sound in its present form?

Yes

Are the interpretations and conclusions justified by the results?

Yes

Is the language acceptable?

Yes

Is it clear how to access all supporting data?

No

Do you have any ethical concerns with this paper?

No

Have you any concerns about statistical analyses in this paper?

No

Recommendation?

Accept with minor revision (please list in comments)

Comments to the Author(s)

A few small comments

Page 2, line 23. Alviniconcha reaches a size of at least 80 mm

Page 3, line 38. Strength of bleach. Do you mean commercial bleach diluted with 4 parts of water?

Then one needs to know the strength of the commercial solution, which varies.

Page 5 line 13. A small discrepancy between frozen at -20° and page 3 line 30, where it says -80°.

Page 11, line 12. "spines in that species is much" should be "are much".

Page 12 line 49. I think it should be emphasized that the total number of specimens, records and localities for the genus is comparatively small which makes zoogeography difficult.

Decision letter (RSOS-190393.R0)

19-Jun-2019

Dear Dr Chen

On behalf of the Editors, I am pleased to inform you that your Manuscript RSOS-190393 entitled "Four new deep-sea provannid snails (Gastropoda: Aabysochrysoidea) discovered from hydrocarbon seep and hydrothermal vents in Japan" has been accepted for publication in Royal Society Open Science subject to minor revision in accordance with the referee suggestions. Please find the referees' comments at the end of this email.

The reviewers and handling editors have recommended publication, but also suggest some minor revisions to your manuscript. Therefore, I invite you to respond to the comments and revise your manuscript.

- Ethics statement

- Data accessibility

It is a condition of publication that all supporting data are made available either as supplementary information or preferably in a suitable permanent repository. The data accessibility section should state where the article's supporting data can be accessed. This section should also include details, where possible of where to access other relevant research materials such as statistical tools, protocols, software etc can be accessed. If the data has been deposited in an external repository this section should list the database, accession number and link to the DOI for all data from the article that has been made publicly available. Data sets that have been

deposited in an external repository and have a DOI should also be appropriately cited in the manuscript and included in the reference list.

If you wish to submit your supporting data or code to Dryad (<http://datadryad.org/>), or modify your current submission to dryad, please use the following link:
<http://datadryad.org/submit?journalID=RSOS&manu=RSOS-190393>

- **Competing interests**

- **Authors' contributions**

- **Acknowledgements**

- **Funding statement**

Because the schedule for publication is very tight, it is a condition of publication that you submit the revised version of your manuscript before 28-Jun-2019. Please note that the revision deadline will expire at 00.00am on this date. If you do not think you will be able to meet this date please let me know immediately.

When submitting your revised manuscript, you will be able to respond to the comments made by

the referees and upload a file "Response to Referees" in "Section 6 - File Upload". You can use this to document any changes you make to the original manuscript. In order to expedite the processing of the revised manuscript, please be as specific as possible in your response to the referees. We strongly recommend uploading two versions of your revised manuscript:

If your manuscript is newly submitted and subsequently accepted for publication, you will be asked to pay the article processing charge, unless you request a waiver and this is approved by Royal Society Publishing. You can find out more about the charges at <http://rsos.royalsocietypublishing.org/page/charges>. Should you have any queries, please contact opscience@royalsociety.org.

Kind regards,
Alice Power

Editorial Coordinator
Royal Society Open Science
openscience@royalsociety.org

on behalf of Dr Punidan Jeyasingh (Associate Editor) and Kevin Padian (Subject Editor)
openscience@royalsociety.org

Associate Editor Comments to Author (Dr Punidan Jeyasingh):

This manuscript describes four new species of snails from deep-sea ecosystems. This is a clearly written manuscript with obvious importance. The manuscript was reviewed by an expert who was largely happy with it, although concerns about sample size and biogeography were raised. With much gratitude to the expert reviewer, I invite the authors to address these issues by providing explicit detail, and acknowledging limitations.

Reviewer comments to Author:
Reviewer: 1

Comments to the Author(s)

A few small comments

Page 2, line 23. Alviniconcha reaches a size of at least 80 mm

Page 3, line 38. Strength of bleach. Do you mean commercial bleach diluted with 4 parts of water?

Then one needs to know the strength of the commercial solution, which varies.

Page 5 line 13. A small discrepancy between frozen at -20° and page 3 line 30, where it says -80°.

Page 11, line 12. "spines in that species is much" should be "are much".

Page 12 line 49. I think it should be emphasized that the total number of specimens, records and localities for the genus is comparatively small which makes zoogeography difficult.

Author's Response to Decision Letter for (RSOS-190393.R0)

See Appendix A.

Decision letter (RSOS-190393.R1)

27-Jun-2019

Dear Dr Chen:

On behalf of the Editors, I am pleased to inform you that your Manuscript RSOS-190393.R1 entitled "Four new deep-sea provannid snails (Gastropoda: Abysochrysoidea) discovered from hydrocarbon seep and hydrothermal vents in Japan" has been accepted for publication in Royal Society Open Science subject to minor revision in accordance with the referee suggestions. Please find the referees' comments at the end of this email.

The reviewers and Subject Editor have recommended publication, but also suggest some minor

revisions to your manuscript. As you will see, the Editors have asked that you check the ZooBank links – if these links are not ‘live’ until acceptance of the paper, you may like to double-check the URLs to confirm they’re correct, and this decision gives you an opportunity to do a last check of your text for any minor typographical errors (though none have been noted by the Editors). We also note that your GenBank accession IDs are not yet live; we do recommend that your GenBank record is now public. When resubmitting your manuscript, as a general request, we ask you to please conduct final check of the following:

- Ethics statement

- Data accessibility

If you wish to submit your supporting data or code to Dryad (<http://datadryad.org/>), or modify your current submission to dryad, please use the following link:
<http://datadryad.org/submit?journalID=RSOS&manu=RSOS-190393.R1>

- Competing interests

- Authors’ contributions

All submissions, other than those with a single author, must include an Authors’ Contributions section which individually lists the specific contribution of each author. The list of Authors should meet all of the following criteria; 1) substantial contributions to conception and design, or acquisition of data, or analysis and interpretation of data; 2) drafting the article or revising it critically for important intellectual content; and 3) final approval of the version to be published.

- Acknowledgements

- Funding statement

Because the schedule for publication is very tight, it is a condition of publication that you submit the revised version of your manuscript before 06-Jul-2019. Please note that the revision deadline will expire at 00.00am on this date. If you do not think you will be able to meet this date please let me know immediately.

Kind regards,
Lianne Parkhouse
Editorial Coordinator
Royal Society Open Science
openscience@royalsociety.org

on behalf of Dr Punidan Jeyasingh (Associate Editor) and Kevin Padian (Subject Editor)
openscience@royalsociety.org

Associate Editor Comments to Author (Dr Punidan Jeyasingh):

I thank the authors for addressing reviewer comments and turning this around quickly. I am happy with it. One remaining issue: Please check links for ZooBank - they are not working.

Author's Response to Decision Letter for (RSOS-190393.R1)

See Appendix B.

Decision letter (RSOS-190393.R2)

04-Jul-2019

Dear Dr Chen,

I am pleased to inform you that your manuscript entitled "Four new deep-sea provannid snails (Gastropoda: Aabysochrysoidea) discovered from hydrocarbon seep and hydrothermal vents in Japan" is now accepted for publication in Royal Society Open Science.

on behalf of Dr Punidan Jeyasingh (Associate Editor) and Kevin Padian (Subject Editor)
openscience@royalsociety.org

Appendix A

Dear Dr. Jeyasingh,

I am pleased to submit to *Royal Society Open Science* the revised version of the Research Article manuscript “**Four new deep-sea provannid snails (Gastropoda: Abysochrysoidea) discovered from hydrocarbon seep and hydrothermal vents in Japan**” (Manuscript ID RSOS-190393).

We appreciated the constructive criticisms and comments from the reviewer, and we thank you for providing this opportunity for us to improve this manuscript and submit a revised version.

A point-by-point response to comments is included below. We have also provided the necessary museum accession numbers for the type specimens.

We look forward to seeing the paper published in *Royal Society Open Science*.

Best regards,

Chong Chen
Research Scientist
JAMSTEC

RESPONSE TO THE HANDLING EDITOR (Dr Punidan Jeyasingh)

This manuscript describes four new species of snails from deep-sea ecosystems. This is a clearly written manuscript with obvious importance. The manuscript was reviewed by an expert who was largely happy with it, although concerns about sample size and biogeography were raised. With much gratitude to the expert reviewer, I invite the authors to address these issues by providing explicit detail, and acknowledging limitations.

Thank you very much! We have improved the manuscript according to the suggestions by the reviewer, as follows, and we hope the present version is acceptable for publication.

RESPONSE TO REVIEWER 1

A few small comments

We are grateful for your constructive review of our manuscript, and we are glad that you are positive about its publication.

Page 2, line 23. Alviniconcha reaches a size of at least 80 mm

Corrected to '*over 80 mm in maximum shell height*'.

Page 3, line 38. Strength of bleach. Do you mean commercial bleach diluted with 4 parts of water? Then one needs to know the strength of the commercial solution, which varies.

Yes this is correct – and we can agree about the strength of the solutions should be stated.

We have therefore modified this sentence as follows:

'Any remaining soft tissue was cleaned using commercial bleach (concentration of sodium hypochlorite: 5%) diluted with four parts of water for two to five minutes until complete dissolution.'

Page 5 line 13. A small discrepancy between frozen at -20° and page 3 line 30, where it says -80°.

Thanks for spotting this. This should be -20°C. We have corrected the mistake on page 3 line 30.

Page 11, line 12. "spines in that species is much" should be "are much".

Corrected.

Page 12 line 49. I think it should be emphasized that the total number of specimens, records and localities for the genus is comparatively small which makes zoogeography difficult.

Many thanks, this is a point well-worth stating. We have added a sentence in this paragraph to emphasise this idea:

'This also indicates that the total number of specimens, records, and discovered localities of the genus is still comparatively small, which makes a comprehensive understanding of its zoogeography difficult.'

Appendix B

RESPONSE TO THE HANDLING EDITOR (Dr Punidan Jeyasingh)

I thank the authors for addressing reviewer comments and turning this around quickly. I am happy with it. One remaining issue: Please check links for ZooBank - they are not working.

Many thanks, that's great. The **ZooBank links now made public** and they are working. The **GenBank numbers/data are also now public and live.**

I will be grateful if you could accept the paper formally.

Thanks again!

Best,
Chong